

# Lattice regularization of reduced Kähler-Dirac fermions and connections to chiral fermions

Simon Catterall⋆

Department of Physics, Syracuse University, Syracuse, NY 13244, USA

⋆ smcatter@syr.edu

## Abstract

We show how a path integral for reduced Kähler-Dirac fermions suffers from a phase ambiguity associated with the fermion measure that is an analog of the measure problem seen for chiral fermions. However, unlike the case of chiral symmetry, a doubler free lattice action exists which is invariant under the corresponding onsite symmetry. This allows for a clear diagnosis and solution to the problem using mirror fermions resulting in a unique gauge invariant measure. By introducing an appropriate set of Yukawa interactions which are consistent with 't Hooft anomaly cancellation we conjecture the mirrors can be decoupled from low energy physics. Moreover, the minimal such Kähler-Dirac mirror model yields a light sector which corresponds, in the flat space continuum limit, to the Pati-Salam GUT model.



# 1 Chiral symmetry, Kähler-Dirac fermions and the lattice

Efforts to give a non-perturbative definition of a chiral gauge theory go back to the earliest days of lattice field theory. Unfortunately the Nielsen-Ninomiya theorem puts strong constraints on attempts to construct theories with chiral lattice fermions [1]. In essence it says that any lattice theory with an exact $\gamma_5$ symmetry will necessarily contain equal numbers of left and right handed fields. This is naturally understood on the basis of anomalies - the exact chiral symmetry of the lattice theory should naively survive quantization and hence cannot generate the required chiral anomaly in the continuum limit. The lattice achieves this by automatically generating a vector-like theory via the well known phenomenon of fermion doubling.

The discovery of domain wall and overlap fermions [2–6] led to doubler free lattice actions that are invariant under a modified and mildly non-local lattice chiral symmetry - see [7] for recent work on these constructions. The nature of this lattice symmetry is such that it is capable of producing the correct anomaly for the global chiral symmetry of Dirac fields. However, because the associated chiral projector is non-local and depends on the background gauge field the associated path integral for a Weyl field picks up a phase depending on the gauge field. Furthermore this phase is not uniquely determined – it can be changed by performing a unitary rotation of the orthonormal bases for the chiral fermions - this is the well known measure problem [8] which renders the theory ill-defined unless a prescription is given to determine this phase. While such a prescription can be shown to exist for (anomaly free) abelian theories there is currently no non-perturbative construction for anomaly free non-abelian theories [9].

One strategy to get around this measure problem involves building a mirror model which starts from the target chiral theory and adds mirror fermions with opposite chirality. The resultant theory, being vector-like, possesses a well defined measure. One then attempts to give masses to the mirror fermions without disturbing the symmetries and dynamics of the target light chiral sector. Many efforts to realize such mirror models using different types of lattice fermion have been tried - see [10, 11] for a review and a discussion of the most recent and best motivated effort using overlap fermions or domain wall fermions [12]. However, they have so far not yielded positive results.[1]

In this paper we describe an analog of this problem in the case where the Dirac equation is replaced by the Kähler-Dirac equation [17]. The relevant symmetry is no longer generated by $\gamma_5$ but by a new operator $\Gamma$ that is unique to Kähler-Dirac fermions [18]. The analog of a chiral fermion is a so-called *reduced* Kähler-Dirac (RKD) field. However, the crucial difference between chiral and RKD fermions is that a doubler free lattice action can be constructed which is invariant under precisely the same $U(1)$ symmetry as the continuum theory. Indeed, this lattice theory can reproduce *exactly* the analog of the axial anomaly seen for chiral fermions. The anomaly arises from the variation of the fermion measure just as in the continuum and captures the non-trivial topology of the background space [19, 20]. It is thus a mixed $U(1)$-gravitational anomaly. These features allow for the construction of a lattice action for RDK fermions where the measure problem can be made explicit and solved using a mirror fermion construction. The additional issue of anomaly cancellation can also be stated precisely and solved even on the lattice.

In addition, we propose a set Yukawa interactions that respect the 't Hooft anomaly cancellation conditions and should be capable of driving the mirrors into a symmetric gapped phase without breaking symmetries of the light sector.

Finally, we point out that the minimal, anomaly free Kähler-Dirac mirror model yields a theory with the global symmetries and matter representations of the Pati-Salam GUT [21]

---

[1]The exception to this seems to be abelian theories in two dimensions where appropriate six fermion interactions have been shown to successfully decouple the mirror states leaving a low energy chiral theory [13–15]. See also the recent construction in [16].

in the naive flat space continuum limit suggesting that lattice Kähler-Dirac constructions are capable of regulating at least a class of chiral gauge theories. In our paper we will mostly focus on four Euclidean dimensions but our analysis can be easily extended to any arbitrary (even) dimension.

## 2  Kähler-Dirac fermions on and off the lattice

The $D$-dimensional continuum Kähler-Dirac action can be written as

$$S = \left[ \overline{\Phi}, (K + m) \Phi \right], \tag{1}$$

where $K = d - d^\dagger$ is a natural square root of the Laplacian and hence a candidate for a fermion operator. The corresponding fermion field $\Phi$ comprises a set of Grassman valued antisymmetric tensor fields ($p$-forms)

$$\Phi = (\phi, \phi_\mu, \phi_{\mu_1 \mu_2}, \dots, \phi_{\mu_1 \dots \mu_p}, \dots, \phi_{\mu_1 \dots \mu_D}). \tag{2}$$

The action of $d$ and $d^\dagger$ on these component forms is given by [22]

$$d\Phi = \left( 0, \partial_\mu \phi, \partial_{\mu_1} \phi_{\mu_2} - \partial_{\mu_2} \phi_{\mu_1}, \dots, \sum_{perms\, \pi} (-1)^\pi \partial_{\mu_1} \phi_{\mu_2 \dots \mu_D} \right), \tag{3}$$

$$-d^\dagger \Phi = \left( \phi^\nu, \phi^\nu_\mu, \dots, \phi^\nu_{\mu_1, \dots, \mu_{D-1}}, 0 \right)_{;\nu}, \tag{4}$$

and the inner product between two Kähler-Dirac fields $A$ and $B$ is defined as

$$[A, B] = \int d^D x \sqrt{g} \sum_p \frac{1}{p!} a^{\mu_1 \dots \mu_D} b_{\mu_1 \dots \mu_D}. \tag{5}$$

In dimension $D$ the field $\Phi$ contains $2^D$ independent components. In a flat space of even dimensions the Kähler-Dirac equation describes the propagation of $2^{D/2}$ degenerate Dirac fields. To see this first construct matrices $\Psi$ and $\overline{\Psi}$ from elements of the Clifford algebra of $D$-dimensional (Euclidean) Dirac matrices using the p-form components of $\Phi$ as coefficents:

$$\Psi = \sum_p \phi_{\mu_1 \dots \mu_p} \gamma^{\mu_1} \cdots \gamma^{\mu_p},$$

$$\overline{\Psi} = \sum_p \overline{\phi}_{\mu_1 \dots \mu_p} (\gamma^{\mu_1} \cdots \gamma^{\mu_p})^\dagger. \tag{6}$$

It can be easily shown that the original Kähler-Dirac action given in eqn. 1 can then be written as

$$S = \int d^D x \sum_{a=1}^{2^{D/2}} \overline{\Psi}^a \left( \gamma^\mu \partial_\mu + m \right) \Psi^a, \tag{7}$$

where the index $a$ labels columns of the matrix $\Psi$ each of which corresponds to a Dirac spinor.[2] This equivalence between a Kähler-Dirac field and a multiplet of degenerate Dirac fields is no longer true in a curved space as the gravitational coupling of tensors is very different from

---

[2]In odd dimensions the mapping to spinors is more subtle – the matrix representation of the $D = 2n + 1$ dimensional Kähler-Dirac theory requires employing gamma matrices of size $2^{n+1}$ [23]. The degrees of freedom carried by the resultant matrix fermion are then reduced by a factor of two using the projectors described in section 3.

spinors. Indeed this observation lies at the heart of why there are new anomalies of Kähler-Dirac fermions that are different from the usual anomalies of Dirac fermions. In particular Kähler-Dirac fermions are allowed on non spin manifolds and do not require the additional structure of frames and spin connections. One obvious manifestation of this difference is the observation that the Kähler-Dirac equation possesses zero modes even on spaces with positive curvature such as the sphere $S^D$.

The flat space action in eqn. 7 is invariant under both (Euclidean) $SO(4)$ Lorentz symmetry and an $SU(2^{D/2})$ flavor symmetry.

$$\Psi \to L\Psi F^\dagger. \tag{8}$$

However the original Kähler-Dirac theory is invariant only under a diagonal subgroup of these symmetries corresponding to the matrix transformation

$$\Psi \to L\Psi L^\dagger. \tag{9}$$

This transformation makes it clear how the Lorentz transformation properties of a set of spinors can be compatible with a collection of tensors.[3] One simply employs the relation $L\gamma_\mu L^\dagger = \Lambda_\mu^\nu \gamma_\nu$ where $\Lambda_\mu^\nu$ is the usual boost matrix for vectors together with the expansion of $\Psi$ given in eqn. 6.

We will be particularly interested in an operator $\Gamma$ whose action on $p$-form fields is given by

$$\Gamma \phi_{\mu_1 \dots \mu_p} = (-1)^p \, \phi_{\mu_1 \dots \mu_p}. \tag{10}$$

It is easy to see that $\Gamma$ anticommutes with $K$. This allows us to construct a $U(1)$ transformation

$$\Phi \to e^{i\alpha\Gamma}\Phi,$$
$$\overline{\Phi} \to \overline{\Phi}e^{i\alpha\Gamma}. \tag{11}$$

This is a symmetry of the massless action but we will show that this is not a symmetry of the theory in curved space due to an anomaly. In the matrix representation it can be realized as

$$\Psi \xrightarrow{\Gamma} \gamma^5 \Psi \gamma^5, \tag{12}$$

and in this context it is sometimes referred to as a twisted chiral symmetry.

One of the prime advantages of the Kähler-Dirac equation over the Dirac equation is that it can be formulated on a discrete approximation to a continuum space such as a triangulation [24]. The $p$-form fields are to be replaced by $p$-cochains – lattice fields defined on $p$-simplices. Each $p$-simplex is specified by a list of $(p+1)$ vertex labels $[a_0, \dots a_p]$ and the p-simplex field $\Phi^{(p)}$ is given by the formal sum

$$\Phi^{(p)} = \sum_{p-\text{simplices}} [a_0, \dots, a_p] \phi^{(p)}_{[a_0, \dots, a_p]}. \tag{13}$$

Discrete analogs of $d$ and its adjoint $d^\dagger$ exist - the co-boundary $\bar{\delta}$ and boundary $\delta$ operators [19, 20, 24, 25] with the action of $\delta$ on a p-simplex field being given by

$$\delta_p \Phi^{(p)} = \sum_{k=0}^p (-1)^k [a_0, \dots \hat{a}_k, \dots, a_p] \phi^{(p-1)}_{[a_0, \dots \hat{a}_k, \dots, a_p]}, \tag{14}$$

where $\hat{a}_k$ denotes the vertex that is removed to get the $k^{\text{th}}$ boundary $(p-1)$-simplex. It is easy to see that $\delta^2 = (\delta^\dagger)^2 = 0$ just as for the continuum exterior derivative. Using the abbreviated notation

$$C_p \equiv [a_0, \dots, a_p], \tag{15}$$

---

[3]Kähler-Dirac fermions are most naturally formulated in Euclidean signature [22]. Extensions to Minkowski signature are possible but typically lead to non-compact flavor symmetries.

we can write this as

$$\delta_p \phi(C_p) = \sum_{C_{p-1}} I(C_p, C_{p-1}) \phi(C_{p-1}), \tag{16}$$

where $I(C_p, C_{p-1})$ is a $N_p \times N_{p-1}$ incidence matrix whose matrix elements are $+1$ if $C_{p-1}$ is contained in the boundary of $C_p$ with the correct orientation, $-1$ if it occurs with opposite orientation and zero otherwise.

Furthermore, a natural analog of the continuum Kähler-Dirac action exists for such a triangulation

$$S = \left[ \overline{\Phi}, (\delta - \delta^\dagger + m) \Phi \right], \tag{17}$$

where $\Phi = \sum_{p=0}^{D} \Phi^{(p)}$ is the set of all p-cochains in the triangulation, $\delta = \sum_p \delta_p$ and the inner product now involves a sum over the triangulation of products of p-cochain elements. Crucially, the number of zero modes of the continuum theory, being related to the ranks of the homology groups, are reproduced exactly in the discrete theory. This guarantees that no additional fermionic species arise in the lattice theory as compared to the continuum.

It is also important to recognize that the properties of the operator $\Gamma$ and hence the classical $U(1)$ symmetry survive intact under discretization. Much of our discussion will focus on this discrete case where the Kähler-Dirac operator becomes a finite dimensional matrix. This is a key difference over efforts to treat chiral fermions on the lattice where even the most promising efforts require some degree of non-locality in the lattice realization of $\gamma_5$ or the fermion operator.

## 3 Reduced Kähler-Dirac fermions and a measure problem

Using the operator $\Gamma$ we can define the projectors $P_\pm = \frac{1}{2}(1 \pm \Gamma)$ and hence decompose a Kähler-Dirac field into two independent *reduced* fields $\Phi_+ = P_+ \Phi$ and $\Phi_- = P_- \Phi$ with $\Gamma = +1$ and $\Gamma = -1$ respectively. This is analogous to the decomposition of a Dirac fermion into left and right handed Weyl fermions. Notice it can be done in the discrete theory with the continuum $\Gamma$ operator - it does not require the introduction of a modified, non-local $\hat{\gamma}_5$ operator unlike in overlap fermion constructions of lattice Weyl fermions. Using these projectors it is then easy to see that the *massless* action breaks into two independent pieces

$$S = \left[ \overline{\Phi}_-, K \Phi_+ \right] + \left[ \overline{\Phi}_+, K \Phi_- \right]. \tag{18}$$

Thus the fields $\Phi_+$ and $\Phi_-$ contain only even or odd p-forms or cochains respectively. The action for a single *reduced* Kähler-Dirac fermion with say $\Gamma = +1$ is gotten by dropping one of these pieces, for example

$$S_{\text{RKD}} = \left[ \overline{\Phi}_-, K \Phi_+ \right] = \left[ \overline{\Phi}_-, \mathcal{K} \Phi_+ \right], \quad \text{with} \quad \mathcal{K} = P_- K P_+. \tag{19}$$

Focusing on the discrete case, it is important to notice that in even dimensions, the operator $\mathcal{K}$ generically corresponds to a rectangular matrix since the number of even and odd simplices are not equal if the space is not flat. This can be seen explicitly from the Euler relation

$$\chi = N_0 - N_1 + N_2 - N_3 + \ldots + N_D = N_+ - N_-, \tag{20}$$

where $N_\pm$ denote the numbers of even and odd dimensional simplices. Thus, if one tries to integrate out the reduced fermions in a triangulation with non-zero $\chi$ one immediately encounters a problem since one cannot even define $\det(\mathcal{K})$. The way to proceed is to use singular value decomposition (SVD) to write

$$\mathcal{K} = U \Sigma V^\dagger, \tag{21}$$

where $U$ is a $N_- \times N_-$ unitary matrix, $V$ is a $N_+ \times N_+$ unitary matrix and $\Sigma$ is a diagonal rectangular matrix with real, positive semi-definite singular values $\sigma^i$, $i = 1 \ldots \min(N_+, N_-)$. The columns of $U$ define an orthonormal basis $\{u^i, i = 1 \ldots N_-\}$ of the odd cochains and can be computed as the eigenvectors of $\mathcal{K}\mathcal{K}^\dagger$ while the columns of $V$ define a basis $\{v^i, i = 1 \ldots N_+\}$ for the even cochains and correspond to the eigenvectors of $\mathcal{K}^\dagger\mathcal{K}$. The singular values $\sigma^i$ can be obtained as the (positive) square roots of the non-zero eigenvalues of either $\mathcal{K}\mathcal{K}^\dagger$ or $\mathcal{K}^\dagger\mathcal{K}$. The matrix $\mathcal{K}$ maps between the $i$th column vector of $V$ and the corresponding column vector of $U$.

$$\mathcal{K}v^i = \sigma^i u^i, \quad i = 1 \ldots \min(N_+, N_-). \tag{22}$$

For the remaining discussion we will focus on the case of a background space with spherical topology with $\chi = 2$ which corresponds to a natural compactification of flat space and yields $N_+ = N_- + 2$. Using these results one can write down an expression for the partition function of such a reduced Kähler-Dirac system by setting

$$\overline{\Psi}_- = \overline{\Phi}_- U,$$
$$\Psi_+ = V^\dagger \Phi_+,$$
$$D\overline{\Phi}_- D\Phi_+ = D\overline{\Psi}_- D\Psi_+ \left[ \det(U^\dagger) \det(V) \right]. \tag{23}$$

Integrating out the fermions ignoring any zero modes yields

$$Z = \det(U^\dagger) \det(V) \prod_i^{N_-} \sigma_i, \tag{24}$$

where $\prod_{i=1}^{N_-} \sigma_i = \sqrt{\det(\mathcal{K}\mathcal{K}^\dagger)}$ corresponds to the positive square root of the full Kähler-Dirac operator omitting zero modes. Clearly the partition function acquires a phase from the product of the two determinants. What is worse however is that this phase is not unique. In fact the matrices $U$ and $V$ are not uniquely determined by the original fermion operator - any column vector of $V$ can be multiplied by a phase while preserving the orthonormal character of the basis. Because of the mapping given in eqn. 22 the corresponding column of $U$ must be multiplied by the same phase. These phases then cancel between $\det(U^\dagger)$ and $\det(V)$ *except* for any phase associated with the two unpaired columns of $V$ that correspond to zero modes of $\mathcal{K}$.

In fact it should be clear there remains an ambiguity in the phase of fermion measure for any $\chi \neq 0$. It is analogous to a similar phase ambiguity seen in the construction of a path integral for chiral fermions [8]. In the free theory this extra phase can be held fixed and causes no real difficulty since it cancels out in any observable.

However, this problem becomes more severe in the situation where one tries to gauge $U(1)$ symmetry given in eqn. 11. To gauge the action one simply replaces the incidence matrices $I(C_p, C_{p-1})$ by $U(1)$-valued gauge fields $W(C_p, C_{p-1})$ that transform according to

$$W(C_p, C_{p-1}) \rightarrow \omega^\dagger(C_p) W(C_p, C_{p-1}) \omega^\dagger(C_{p-1}), \tag{25}$$

where $\omega(C_p) = e^{i\alpha(C_p)\Gamma} = e^{i\alpha(C_p)(-1)^p}$ varies for each element of the $p$-cochain [26]. In this case the determinants of the unitary matrices $U$ and $V$ depend on the gauge field $W$ and hence the partition function acquires a gauge field dependent phase

$$Z = e^{iA(W)} \sqrt{\det(\mathcal{K}\mathcal{K}^\dagger)}. \tag{26}$$

We can then ask whether the effective action $A(W)$ is gauge invariant. Under a gauge transformation

$$\mathcal{K} \rightarrow \omega_-^\dagger \mathcal{K} \omega_+^\dagger, \tag{27}$$

where we have explicitly labeled the gauge rotation $\omega(C_p)$ by whether is applied to even or odd co-chains. Under this transformation the matrix $U \to \omega_-^\dagger U$ and $V \to \omega_+ V$ where we view each $\omega_\pm$ as a diagonal matrix of phases. The fermion measure will hence pick up an additional phase factor

$$\det(\omega_+)\det(\omega_-). \tag{28}$$

Clearly, the imaginary part of the effective action $A(W)$ is not even gauge invariant and hence the theory of a single reduced Kähler-Dirac fermion suffers from a gauge anomaly which ruins the consistency of the theory. It is completely analogous to the usual $U(1)$ chiral anomaly of a single Weyl field.[4]

One can imagine restoring gauge invariance by considering a set of reduced fields with differing charges $q_a$ under the $U(1)$ symmetry. The determinant factors are replaced by

$$\det\left(\omega_+^{\sum_a q_a}\right) \times \det\left(\omega_-^{\sum_a q_a}\right). \tag{29}$$

Clearly a set of fields satisfying $\sum_a q_a = 0$ will yield a gauge invariant phase $A(W)$.

However, even in the case where the gauge anomaly is cancelled, it still suffers from the ambiguity discussed earlier – on the sphere one can still multiply two columns of $V$ by arbitrary phases which can now vary with the background gauge field $W$. This would shift $A(W) \to A(W) + \delta(W)$ for arbitrary $\delta(W)$. This is the fermion measure problem for reduced Kähler-Dirac fields. Without a prescription for determining $\delta(W)$ the theory is simply ill-defined. It should be contrasted with the measure problem for chiral fermions where the phase ambiguity arises both from zero modes and additionally the non-local nature of the projector $\hat{\gamma}_5$.

To address this larger issue one can instead add a set of mirror fermions with the same charges and symmetries as the original reduced fermions but with the opposite eigenvalue of $\Gamma$. For simplicity consider a single reduced field with $\Gamma = +1$ and its mirror with $\Gamma = -1$. The resultant (anomaly free) fermion operator will correspond to a direct sum of $N_- \times N_+$ and $N_+ \times N_-$ matrices for the original and mirror sectors respectively which can be embedded into a square matrix of size $(N_+ + N_-) \times (N_+ + N_-)$. Because of differences in the dynamics of the light and mirror fermions, the SVD of this matrix will generically lead to a non-zero phase but the latter will now be well defined and gauge invariant - the measure problem will be resolved.[5]

Of course if this system is to reproduce a theory of reduced fermions with fixed $\Gamma$ eigenvalue at low energy one will need to find a suitable set of mirror interactions which are capable of gapping all the mirror states without affecting the original fermions. We now turn to this problem.

## 4 The minimal anomaly free mirror model

The simplest mirror model that one can conceive of corresponds to adding a mirror field $\Phi_-$ with $\Gamma = -1$. The corresponding mirror action takes the form

$$S_{\text{mirror}} = \overline{\Phi}_+ \hat{\mathcal{K}} \Phi_-, \tag{30}$$

where $\hat{\mathcal{K}} = P_+ K P_- = -\mathcal{K}^\dagger$. The combined system is described by a square matrix of size $N_+ + N_-$ corresponding to a full Kähler-Dirac field and the measure is now well defined.

Actually, even in this case, the theory suffers from a residual problem which needs to be addressed. Consider the limit where the $U(1)$ gauge coupling is sent to zero and focus on the

---

[4]The omission of the zero modes also introduces a non-local constraint into the fermion measure.

[5]Although the theory will still suffer from a sign problem.

global $U(1)$ symmetry. It is straightforward to see that this symmetry is in fact anomalous by considering the case where the gauge rotation $\omega = e^{i\alpha}$ is taken to be a constant. Then, the change in the phase of $Z$ is just given by

$$Z \rightarrow e^{2i\alpha(N_+ - N_-)}Z = e^{2i\chi\alpha}Z , \tag{31}$$

where the factor of two in the exponent accounts for the presence of two reduced fermions. On a background with the topology of the sphere we hence see that the global $U(1)$ symmetry is broken to $Z_4$ [19].

Thus the symmetry we were were considering in section 3 is not $U(1)$ but only $Z_4$ on a curved space with the topology of the sphere. If we imagine gauging this $Z_4$ the anomaly cancellation condition that guarantees gauge invariance of the measure for a system of reduced fermions with $\Gamma = +1$ say is then

$$\sum_a q_a = 0 \mod 4 . \tag{32}$$

It is interesting to note that that four massless reduced Kähler-Dirac fermions in four dimensions can be decomposed into eight Dirac or sixteen Majorana fermions in the flat space continuum limit. This agrees with the critical numbers of Weyl fermions needed to cancel the spin-$Z_4$ anomaly in flat space and also the number of boundary chiral fermions that can be gapped in five dimensional topological superconductors [27, 28]. Indeed this condition, combined with the dimension dependent counting implied by the decomposition of a Kähler-Dirac field into spinors, is capable of predicting the numbers of fermions needed to cancel a variety of discrete 't Hooft anomaly for free Majorana fermions in any dimension [29–33].

Of course if the goal is to construct a theory with a light sector containing only reduced fermions with say $\Gamma = +1$ we will, in addition, need to find a mechanism that is capable of lifting all the mirror states with $\Gamma = -1$ to high energy without disturbing this light sector. A necessary condition for such symmetric mass generation to occur is that the mirror sector must be free of all 't Hooft anomalies - here the mixed $Z_4$-gravitational anomaly just described. Hence, the minimal mirror model of Kähler-Dirac fermions which is free of such 't Hooft anomalies comprises four reduced fields with say $\Gamma = +1$ representing the light fermions and a mirror sector containing four reduced fields with $\Gamma = -1$ (or vice versa). These four flavors of fermion in each sector would then inherit a global $SU(4)$ symmetry.

The simplest dynamics that we can add to a model with four flavors of reduced field that is potentially capable of gapping the mirrors corresponds to coupling the fermions to a scalar field $\phi$ in the six dimensional antisymmetric representation of $SU(4)$. To retain the $Z_4$ symmetry of the model we take these scalars to change sign under a $Z_4$ transformation which is reminiscent of the the action of the spin-$Z_4$ symmetry of chiral fermions [27, 34]. We can rewrite the reduced mirror fields in terms of a doublet

$$\Lambda_{\mathrm{m}} = \begin{pmatrix} \overline{\Phi}_+ \\ \Phi_- \end{pmatrix} , \tag{33}$$

corresponding to the mirror action

$$S_{\mathrm{m}} = \sum \Lambda_m^T M_{\mathrm{m}} \Lambda_m , \tag{34}$$

where the mirror fermion operator $M_{\mathrm{m}}$ has the form

$$M_{\mathrm{m}}^{ab} = \begin{pmatrix} \lambda_{\mathrm{m}}\phi^{ab} & \delta^{ab}\hat{\mathcal{K}} \\ -\delta^{ab}\hat{\mathcal{K}}^T & \lambda_{\mathrm{m}}\hat{\phi}^{ab} \end{pmatrix} , \tag{35}$$

where $\phi$ and $\hat{\phi}$ are $4N_+ \times 4N_+$ and $4N_- \times 4N_-$ matrices of scalars coupled only to the even cochain and odd cochain fields respectively. In fact these Yukawa couplings are needed to soak up the zero modes of $\hat{\mathcal{K}}$ that occur in the even co-chain sector that would otherwise make the partition function vanish [19]. In fact these zero modes are connected to the appearance of the global $U(1)$ anomaly discussed earlier and are implied by the index theorem

$$n_+ - n_- = \chi, \tag{36}$$

where $n_\pm$ is the number of zero modes of $K$ with eigenvalue $\Gamma = \pm 1$ on a manifold with Euler character $\chi$.

To generate a four fermion term one can then add to the action a term that is quadratic in the scalars, but conservation of $Z_4$ actually allows for any term that is even in the number of scalars including, for example, a scalar kinetic term.[6] The hope would be that if the mirror sector coupling $\lambda_m$ is large enough the mirror sector can be driven into a symmetric massive phase corresponding to the presence of a four fermion condensate composed of mirror fermions while leaving the light sector intact - see the review on symmetric mass generation and references therein [36]. Evidence in favor of this scenario can be found in related simulations of reduced staggered fermions in a variety of dimensions [37–43] and in condensed matter physics [44–47].

In fact one should add a similar term also in the light sector to keep the light sector partition function non-zero. Assembling the light sector fields into an analogous doublet

$$\Lambda_l = \begin{pmatrix} \overline{\Phi}_- \\ \Phi_+ \end{pmatrix}. \tag{37}$$

The associated fermion operator for the light sector would take the form

$$M_l^{ab} = \begin{pmatrix} \lambda_l \hat{\phi}^{ab} & \delta^{ab} \mathcal{K} \\ -\delta^{ab} \mathcal{K}^T & \lambda_l \phi \end{pmatrix}. \tag{38}$$

A small bare $\lambda_l$ can soak up the zero modes but should be irrelevant in the continuum limit with the light sector remaining in a massless symmetric phase.

Notice both reduced fermion operators now correspond to square, complex, antisymmetric matrices and hence integration over the fermions yields Pfaffians for both mirror and light sectors. The phases carried by the matrices $U$ and $V$ defined in our earlier SVD analysis are now generated by these Pfaffians. If we have cancelled all anomalies correctly these Pfaffians are free of 't Hooft anomalies but in general the model will still suffer from a sign problem.

## 5 Connection to the Pati-Salam GUT model

It is interesting to ask what is the spinor content of the light sector which contains four copies of say $\Phi_+$ in flat space. The answer is intriguing – in the continuum it is precisely a theory with the global symmetries and matter representations of the Pati-Salam GUT. This is clearly seen if we work in the continuum, use the matrix representation for the field, and choose a chiral basis for the Dirac gamma matrices

$$\gamma^\mu = \begin{pmatrix} 0 & \sigma_\mu \\ \overline{\sigma}_\mu & 0 \end{pmatrix}, \tag{39}$$

---

[6]Previous numerical work in four dimensions suggests such a term may be needed [35].

with $\sigma_\mu = (I, i\sigma_i)$ and $\overline{\sigma}_\mu = (I, -i\sigma_i)$ Thus

$$\Psi_+^a = \begin{pmatrix} \psi_R^a & 0 \\ 0 & \psi_L^a \end{pmatrix}, \tag{40}$$

where the index $a = 1 \dots 4$ runs over the four copies needed for anomaly cancellation and we assume that the fermions transform in the fundamental representation of a global $SU(4)$ symmetry. The $2 \times 2$ blocks $\psi_L$ and $\psi_R$ are composed of two doublets of Weyl fermions transforming under an "internal" $SU(2) \times SU(2)$ flavor symmetry in addition to $SU(4)$. If we imagine continuing back to Minkowski space this fermion content corresponds to a set of sixteen left-handed Weyl spinors transforming in the representations $(4, 2, 1) \oplus (\overline{4}, 2, 1)$ under the global $SU(4) \times SU(2) \times SU(2)$ symmetry. These are the symmetries and matter representations of the Pati-Salam GUT. It is interesting that the minimal mirror model we construct that cancels off gravitational 't Hooft anomalies for Kähler-Dirac fermions generates a well known chiral gauge theory.[7]

While our discussion has focused on geometries with the topology of the sphere the connection to Pati-Salam is made in flat space. In this limit one can replace the general triangulation by a hypercubic lattice and map the (reduced) lattice Kähler-Dirac field into a (reduced) staggered field [26, 48].

In this case, and assuming we can indeed gap the mirror sector as described, we claim the continuum limit of that staggered fermion theory would correspond to a theory with the global symmetries and matter representations of the Pati-Salam GUT - a chiral gauge theory.

# 6 Conclusions

In this paper we have focused on constructing a path integral describing reduced Kähler-Dirac fermions propagating on a curved background space. This problem is analogous to the problem of defining a path integral for a gauged chiral fermion. However, the Kähler-Dirac case possesses one key advantage over the chiral fermion problem because the operator $\Gamma$ that plays the role of $\gamma_5$ survives intact when the theory is discretized on a triangulation of the space. This allows us to study the problem in a regulated lattice model with an onsite $U(1)$ symmetry and no fermion doubling.[8] The lattice Kähler-Dirac operator for reduced fermions maps the space of even cochains to odd cochains and vice versa. On a space with non-zero Euler characteristic it takes the form of a finite rectangular matrix. To integrate out the fermions then requires the use of singular value decomposition which yields two unitary matrices that determine orthonormal bases in these two spaces. If the Euler character of the triangulation is non-zero there is an ambiguity in the phase of the fermion measure arising from the determinants of these unitary matrices which becomes gauge field dependent after the $U(1)$ symmetry associated to $\Gamma$ is gauged. This fermion measure problem for reduced Kähler-Dirac fermions is very similar to that encountered in defining the path integral for chiral fermions.

As for chiral fermions one can add mirror fermions to remove this phase ambiguity. One then discovers that the global $U(1)$ symmetry of the mirror model is broken via an anomaly to $Z_{2\chi}$. This anomaly can be computed exactly on a finite triangulation contradicting the folklore that only systems with an infinite number of degrees of freedom can generate anomalies. Restricting to the sphere $S^4$ and requiring cancellation of the resulting 't Hooft anomaly one finds that only multiples of four reduced fermions are allowed. In the flat space continuum limit this

---

[7]Early work connecting symmetric mass generation in lattice theories to grand unified theories can be found in [45].

[8]No doubling in this context means that discretization does not produce any additional low energy states that were not already present in the continuum theory.

fermion content corresponds to multiples of sixteen Weyl spinors and our result is in agreement with the cancellation of spin-$Z_4$ anomalies for Weyl fermions. Indeed, the Kähler-Dirac argument can be generalized to any even dimension and yields results for anomaly cancellation for a variety of discrete symmetry [27, 49]. We employ this constraint to identify a set of Yukawa interactions that should be capable of lifting the mirror states to the cut-off without breaking symmetries or disturbing the light sector. There is already numerical evidence that suggests that such symmetric mass generation is possible in such theories.

Remarkably, we show that the light sector of the minimal anomaly free mirror model, in the continuum flat space limit, is nothing more than the Pati-Salam GUT model. Rather surprisingly the search for a non-perturbative construction of anomaly free reduced Kähler-Dirac theories appears to target a class of anomaly free chiral models. Indeed, the mirror model of reduced Kähler-Dirac fields we describe can be thought of as providing a non-perturbative definition of the Pati-Salam model. Indeed, it would seem that any chiral theory with an $SU(2) \times SU(2) \times G$ global symmetry can be handled in this manner as long as $G$ contains an complex, irreducible representation whose dimension is a multiple of four.

Of course to create a true chiral gauge theory one needs to gauge these non-abelian symmetries. Gauging the group $G$ in the lattice theory is straightforward and follows the procedure given in eqn. 11 and the measure is automatically gauge invariant. Gauging the internal $SU(2) \times SU(2)$ symmetry is more problematic. In the case of staggered fermions this symmetry is broken to a $Z_2$ subgroup corresponding to the shift symmetry $\chi(x) \to \chi(x + \mu)\xi_\mu(x)$ where $\chi$ is the staggered fermion field and $\xi_\mu(x) = (-1)^{\sum_{i=\mu+1}^{D} x_i}$. The presence of this exact global $Z_2$ symmetry is enough to ensure restoration of the global $SU(2) \times SU(2)$ symmetry in the continuum limit [50, 51] and one might hope that gauging the $Z_2$ symmetry would allow for an $SU(2) \times SU(2)$ emergent gauge symmetry. However, the offsite nature of the shift symmetry makes it difficult to gauge the $Z_2$. Recent work on Hamiltonian lattice systems with similar shift symmetries may offer a path forward [52, 53].

## Acknowledgments

**Funding information** This work was supported by the US Department of Energy (DOE), Office of Science, Office of High Energy Physics, under Award Number DE-SC0009998.

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
