# Peer review of "Lattice Regularization of Reduced Kähler-Dirac Fermions and Connections to Chiral Fermions"

_SciPost Physics, doi:SciPost Phys. 16, 108 (2024)_

## Round 1 · Referee Report · Anonymous (Referee 1) · 2023-12-14

Strengths

1- the paper is clearly written and well motivated 2-the problem with the ambiguous phase is new and interesting 3-the proposed resolution, to use mirror fermions and attempt decoupling is interesting 4-opens a new arena of future research

Weaknesses

1-the only thing I can criticize is that it requires some background knowledge

Report

I think this is a very interesting paper. I was not aware of the phase issues in theories with gauged reduced Kahler Dirac fermions. The proposal is interesting and worthy of further pursuit. Thus, I recommend publication.

---

## Round 1 · Referee Report · Anonymous (Referee 2) · 2024-2-11

Report

Warnings issued while processing user-supplied markup:

  • Inconsistency: Markdown and reStructuredText syntaxes are mixed. Markdown will be used.
    Add "#coerce:reST" or "#coerce:plain" as the first line of your text to force reStructuredText or no markup.
    You may also contact the helpdesk if the formatting is incorrect and you are unable to edit your text.

The content of this paper has significant overlap with PhysRevD.107.014501 (by the same author). In particular, the following topics are already covered in PhysRevD.107.014501: - the anomaly of the global twisted chiral symmetry of Kähler-Dirac fermions in curved space, - the gauging of the residual Z4 symmetry, - the discussion of a possible mechanism to decouple the mirror sector - the presentation of a model which, assuming that the decoupling mechanism works as expected, reduces to the Pati-Salam model. The discussion in Section 3 is the new contribution of this paper: the author discusses how gauge anomalies are produced in the reduced Kähler-Dirac fermions in generic curved space. In particular, he shows that the underlying mechanism is similar to what happens e.g. with Ginsparg-Wilson fermions in topologically non-trivial gauge fields: the Dirac operator is a rectangular matrix and reasonable attempt to generalise the fermionic determinant produce effective actions that are generically non gauge-invariant and non-local. The discussion is surely interesting and illuminating, and it is worth publication. However three major shortcomings should be addressed.

1- The notation used throughout the paper is confusing and sometimes ambiguous (or at least not completely explained). I notice that the author uses a much more rigorous notation in his PhysRevD.107.014501 paper, and I would recommend to apply the same standards here.

2- Eq. (19) seems problematic to me. The goal of the author is "to gauge the $U(1)$ symmetry given in eq. (8)". The author defines $\omega_p = e^{i \alpha_p \Gamma}$ and states that "$\omega_p$ varies for each element of the $p$-cochain. If this is the case, the notation is confusing since $\omega_p$ suggests a dependence on the order of the cochain only. If I interpret correctly, the gauge transformation is really meant to be defined as

$$ \omega(C_p) = e^{i \alpha(C_p) (-1)^p} $$
where I have used the observation that $\Gamma$ acts as $(-1)^p$ on cochains of order $p$. More explicitly, I guess this means that the fields $\Phi$ and $\bar{\Phi}$ (with a generic charge $q$) are supposed to transform like
$$ \Phi^\omega(C_p) = e^{i q \alpha(C_p) (-1)^p} \Phi(C_p) $$
$$ \bar{\Phi}^\omega(C_p) = \bar{\Phi}(C_p) e^{i q \alpha(C_p) (-1)^p} $$
These equations reduce to eq. 8 if $q=1$ and $\alpha$ is constant. This observation makes me think that I am on the right track. Now, if I interpret eq. (19) as
$$ W(C_p,C_{p-1}) \to \omega(C_p) W(C_p,C_{p-1}) \omega(C_{p-1})^\dagger \ , $$
I get the following explicit expression for the gauge transformed field
$$ W^\omega(C_p,C_{p-1}) = e^{i [ \alpha(C_p) + \alpha(C_{p-1}) ] (-1)^p} W(C_p,C_{p-1}) $$
However this is not consistent with the idea of gauging the $U(1)$ symmetry given in eq. (8): when $\alpha$ is constant, the gauge field $W$ should not transform. It seems to me that the correct transformation should be
$$ W^\omega(C_p,C_{p-1}) = \omega(C_p)^\dagger W(C_p,C_{p-1}) \omega(C_{p-1})^\dagger \ , $$
which can be written explicitly as
$$ W^\omega(C_p,C_{p-1}) = e^{-i [ \alpha(C_p) - \alpha(C_{p-1}) ] (-1)^p} W(C_p,C_{p-1}) $$
This transformation guarantees that the $W$ does not transform under a global transformation. Moreover, it yields
$$ K[W^\omega] \Phi^\omega = \omega^{-q} K[W] \Phi $$
which is what one needs to guarantee invariance of the action. If this is correct, then under a gauge transformation, $K \to \omega^{-q} K \omega^{-q}$ which implies $U \to \omega^{-q} U$ and $V \to \omega^{+q} V$. This is at odds with what stated in the paper (in the case of $q=1$).

I see two possibilities here: either eq. (19) is not correct and should be fixed (together with what comes after), or it is correct but then there is not enough explanation in the paper of why this should be the case.

3- The author fails to comment on the non-locality of eq. (19), which is one the crucial obstructions in the non-perturbative construction of chiral gauge theories. Even when the anomaly cancellation condition is satisfied, the fermionic measure is defined only up to a phase that depends on the gauge field. It is not obvious that this ambiguity can be used to restore the locality of the resulting quantum field theory.

Requested changes

1- In order to make the paper self-contained, it would be good to write explicitly the definitions of the operators $d$ and $d^\dagger$ appearing in eq. (1).

2- The sentence which includes eq. (4) should be clarified: in fact eq. (4) does not follow from anything written above. I guess that the author means something along the lines of: "It can be easily shown that, the equations of motion obtained from the action in eq. (1) are equivalent to:". However, since the paper deals with a quantum theory, my recommendation is to leave out the equations of motion are reformulate the sentence in terms of the action. The sentence may look something like this: "In terms of the variables introduced in (3), the action (eq. 1) can be equivalently written as..."

3- Above eq. (5), it says: " transform only under a diagonal subgroup of the Lorentz and flavor symmetries." The statement is correct, but puzzling at this point, since the flavour symmetry is introduced only after. This could be rearranged in order to improve readability.

4- The transformation given in eq. (8) is show to generate an anomaly in sec. 4. In view of this fact, the statement "This is a symmetry of the massless theory and will be very important in our later discussion of anomalies." is misleading. The author should say clearly that this is a symmetry of the action, but not of the quantum theory in curved space.

5- In general, the formulae would benefit from a more precise notation of the sums. For instance, the sum in eq. (10) is really a sum over all cochains. The sum in eq. (11) is not a sum over the index $p$, but a sum over all (p-1)-cochains. I recommend using $\sum_{C_{p-1}}$ instead of $\sum_{p-1}$. Eq. (12) should be written with a sum rather than an integral. These are just examples, and the notation should be made more precise throughout the paper.

6- Gauge transformations should be denoted by $\omega(C_p)$, instead of $\omega_p$, as they depend on the cochain, not only on its order.

7- Around eq. (19), the symbols $I(C_p,C_{p\pm 1})$ and $W(C_p,C_{p\pm 1})$ are used several times. However $I(C_p,C_{p+1})$ and $W(C_p,C_{p+1})$ are never defined, nor used. So all $\pm$ occurrences should be replaced by $-$.

8- Eq. (19) and the following discussion should be improved, in order to address point 2 in the report.

9- The role of locality should be discussed (see point 3 of the report).

---

## Round 2 · Referee Report · Anonymous · 2024-4-2

Report

The author has address all issues raised in my previous report thoroughly and satisfactorily. I must partially disagree only on one point, i.e. on the issue of locality.

I agree on the fact that both gauge transformation and the fermion operator are strictly local. However, as pointed out by the author, the matrix $\mathcal{K}$ introduced in eq. 19 is rectangular. If one integrates exp(-action) with the local fermionic measure one simply gets zero. In fact the author points out that, in order to obtain the partition function in eq. 24, the zero modes need to be ignored. This meant that the integration measure is defined by restricting the integration domain to the complement of the zero modes, and this operation is non-local (since it cannot be expressed as a local contraint). Therefore, truncating the zero modes, introduced not only a phase-ambiguity and possible breaking of gauge invariance, but also a non-locality.

In a sense this comment does not change the essence of the paper, since the mirror-fermion strategy solves potentially (assuming that the decoupling works) all issues at once, including the non-locality issue. However I believe that this issue should be addresses. In the off-chance that I am mistaken and the operation of throwing away the zero modes can be in fact implemented as a local constraint, then this would be highly non-trivial and should be discussed as well.

---

## Round 2 · Author Response

I have edited the draft to reflect the concerns of referee 3 (referee 2 did not seem to have any specific criticisms). I have added additional material defining the action of $d, d^\dagger$ on the component p-forms that comprise a KD field. I have also fixed the set of typos/clarifications requested by referee 3 -- I would like to thank them for such a careful reading of the manuscript - in general I agreed with all their comments and hope that the new draft is much clearer.

The only real issue I do not agree with is their final
comment concerning eqn 19 (in the previous draft) asking for me to add a discussion on the non-locality associated
with this transformation. This is incorrect - this gauge transformation and indeed the fermion operator are strictly local for the reduced KD fermion - there is no non-locality as appears in say an overlap construction for chiral fermions. There is
only the question of gauge invariance and an ambiguity in the measure associated with the presence of exact zero modes in the lattice KD equation which warrants the
addition of mirror fermions and new dofs. This is emphasized in various parts of the paper.

In the case of overlap fermions there are two sources of measure problems -- the presence of a non-local $\hat{\gamma}_5$ in the projector and the possibility of exact zero modes. In the KD case there is only the latter.

---

## Round 2 · List of Changes

1. Added definitions of $d,d^\dagger$ and additional text to improve the background.
2. Reordered the text as suggested for clarity. Now use the action rather than the equation of motion to discuss symmetries.
3. Again rewritten/reordered together with point 2 for better clarity/logical progression.
4.Fixed.
5. Added new definitions/notation concerning p-cochains and the action of boundary/co-boundary operators
on simplicial lattices throughout to conform to those in PRD paper they cite.
6. Agreed and changed $\omega_p\to \omega(C_p)$ throughout.
7. Agreed. $I(C_p,C_{P+1})$ only arises indirectly since in practice it is gotten by taking the transpose of $I(C_{p+1},C_p)$. I deleted all referencxe therefore to the latter.
8. Corrected eqn19 to agree with their suggestion.
9. See above. Eqn19 does not exhibit any non-locality - this is precisely why (reduced) KD fermions can be handled
in a more complete way than chiral fermions. One one has fixed issues of gauge invariance , remaining measure problem for reduced KD fermions only arises because of the presence of exact zero modes - not because of non-localities in the fermion operator or projector.

---

## Round 3 · Author Response

Hello, added a footnote on page 7 concerning non-locality arising from the presence of zero modes.

---

## Editorial Decision

published